



**Global analysis of continental boundary layer new particle formation based on long-term measurements**

Tuomo Nieminen[1,2], Veli-Matti Kerminen[1], Tuukka Petäjä[1], Pasi P. Aalto[1], Mikhail Arshinov[3], Eija Asmi[4], Urs Baltensperger[5], David C. S. Beddows[6], Johan Paul Beukes[7], Don Collins[8], Aijun Ding[9], Roy M. Harrison[6,10], Bas Henzing[11], Rakesh Hooda[4,12], Min Hu[13], Urmas Hõrrak[14], Niku Kivekäs[4], Kaupo Komsaare[14], Radovan Krejci[15], Adam Kristensson[16], Lauri Laakso[4,7], Ari Laaksonen[4,2], W. Richard Leaitch[17], Heikki Lihavainen[4], Nikolaos Mihalopoulos[18], Zoltán Németh[19], Wei Nie[9], Colin O'Dowd[20], Imre Salma[19], Karine Sellegri[21], Birgitta Svenningsson[16], Erik Swietlicki[16], Peter Tunved[15], Vidmantas Ulevicius[22], Ville Vakkari[4], Marko Vana[14], Alfred Wiedensohler[23], Zhijun Wu[13], Annele Virtanen[2], and Markku Kulmala[1,9,24]

1  Institute for Atmospheric and Earth System Research / Physics, Faculty of Science, University of Helsinki, Finland

2  Department of Applied Physics, University of Eastern Finland, Kuopio, Finland

3  V.E. Zuev Institute of Atmospheric Optics SB RAS, Tomsk, Russia

4  Finnish Meteorological Institute, Helsinki, Finland

5  Laboratory of Atmospheric Chemistry, Paul Scherrer Institute, Villigen, Switzerland

6  School of Geography, Earth and Environmental Sciences, University of Birmingham, Birmingham, UK

7  Unit for Environmental Sciences and Management, North-West University, Potchefstroom, South Africa

8  Department of Atmospheric Sciences, Texas A&M University, USA

9  Joint International Research Laboratory of Atmospheric and Earth System Sciences, School of Atmospheric Sciences, Nanjing University, Nanjing 210023, China

10  Department of Environmental Sciences / Center of Excellence in Environmental Studies, King Abdulaziz University, PO Box 80203, Jeddah, 21589, Saudi Arabia

11  Netherlands Organization for Applied Scientific Research (TNO), Utrecht, the Netherlands

12  The Energy and Resources Institute, IHC, Lodhi Road, New Delhi, India

13  State Key Joint Laboratory of Environmental Simulation and Pollution Control, College of Environmental Sciences and Engineering, Peking University, Beijing 100871, China

14  Institute of Physics, University of Tartu, Tartu, Estonia

15  Department of Environmental Science and Analytical Chemistry & Bolin Centre of Climate Research, Stockholm University, Stockholm, Sweden

16  Department of Physics, Lund University, Lund, Sweden

17  Climate Research Division, Environment and Climate Change Canada, Toronto, Canada

18  Department of Chemistry, University of Crete, Heraklion, Greece

19  Institute of Chemistry, Eötvös University, Budapest, Hungary





[20] School of Physics and Centre for Climate and Air Pollution Studies, National University of Ireland Galway, Ireland

[21] Laboratoire de Météorologie Physique, Observatoire de Physique du Globe de Clermont-Ferrand, Université Clermont-Auvergne, CNRS UMR6016, Aubière, France

[22] Department of Environmental Research, SRI Center for Physical Sciences and Technology, Vilnius, Lithuania

[23] Leibniz Institute for Tropospheric Research, Leipzig, Germany

[24] Aerosol and Haze Laboratory, Beijing Advanced Innovation Center for Soft Matter Science and Engineering, Beijing University of Chemical Technology, Beijing, China

Correspondence to: T. Nieminen (tuomo.nieminen@uef.fi)



**Abstract**

Atmospheric new particle formation (NPF) is an important phenomenon in terms of the global particle number concentrations. Here we investigated the frequency of NPF, formation rates of 10 nm particles and growth rates in the size range of 10–25 nm using at least one year of aerosol number size-distribution observations at 36 different locations around the world. The majority of these measurement sites are in the Northern Hemisphere. We found that the NPF frequency has a strong seasonal variability, taking place on about 30% of the days in March–May and on about 10% of the days in December–February. The median formation rate of 10 nm particles varies by about three orders of magnitude (0.01–10 cm$^{-3}$ s$^{-1}$) and the growth rate by about an order of magnitude (1–10 nm h$^{-1}$). The smallest values of both formation and growth rates were observed at polar sites and the largest ones in urban environments or anthropogenically influenced rural sites. The correlation between the NPF event frequency and the particle formation and growth rate was at best moderate between the different measurement sites, as well as between the sites belonging to a certain environmental regime. For a better understanding of atmospheric NPF and its regional importance, we would need more observational data from different urban areas in practically all  parts of the world, from additional remote and rural locations in Northern America, Asia and most of the Southern Hemisphere (especially Australia), from polar areas, and from at least a few locations over the oceans.

## 1 Introduction

Atmospheric aerosol particles have large impacts on air quality and human health (Apte et al., 2015; Brauer et al., 2015; Lelieveld et al., 2015, Zhang et al., 2015), on the current and future behavior of the climate system (IPCC, 2013; Shindell et al., 2015), and on climate-air quality interactions (Makkonen et al., 2012; Lacressonniere et al., 2014; Pietikäinen et al., 2015; Westerveld et al., 2015; Shen et al., 2017). According to large-scale model simulations, globally the most important source of atmospheric aerosol particles, at least in terms of their total number concentration but perhaps also of climate-relevant particles, is atmospheric new particle formation (NPF) and subsequent particle growth (e.g. Spracklen et al, 2008; Merikanto et al., 2009; Yu et al., 2010; Dunne et al., 2016). The relative importance of atmospheric NPF and primary emissions of aerosol particles into the atmosphere is, however, expected to vary regionally, as well as over the course of the year in any specific location.

Particle number size distribution measurements suggest that atmospheric boundary layer NPF is dominated by regional scale NPF events. These events typically last for at least a few hours and simultaneously take place over distances of hundreds of kilometers. Regional NPF events have been observed worldwide (e.g. Kulmala et al., 2004) and also been characterized for a few relatively large areas in Europe, China and North America (Manninen et al., 2010; Peng et al., 2014; Pietikäinen et al., 2014; Yu et al., 2015; Kulmala et al., 2016; Vana et al., 2016; Berland et al., 2017; Wang et al., 2017). In spite of numerous and an increasing number of high-quality atmospheric aerosol size-distribution measurements, we are still lacking a global, observationally-based and internally-consistent data set on atmospheric NPF that would cover the full annual cycle. Such data, especially from the Southern Hemisphere and tropics, would be valuable for multiple purposes, including global and regional model validation and



complementary use of various modeling and measurement tools to enhance our general understanding of this phenomenon.

The primary goal of this study is to present the first global-scale picture on the main characteristics of atmospheric NPF based on atmospheric observations, including the frequency

of regional NPF events and the formation and growth rates of the newly-formed particles during these events. More specifically, we aim to shed new light on the following questions: (1) how frequent is regional NPF in different types of continental environments overall and during the different seasons?, (2) how do the particle formation and growth rates, as recorded during the observed NPF events, vary with the type of environment and season?, and (3) to which extent

are the NPF event frequency and the particle formation and growth rates connected with each other?

In order to address our goal and specific questions, we gathered observations of atmospheric NPF from several measurement sites where at least one year of particle number size distribution measurements is available. Since the number of such sites turned out to be rather limited, we

included sites with shorter data coverage, provided that these data could be parsed into a full seasonal cycle. The published peer-reviewed articles do not always present NPF event frequencies or particle formation and growth rates. Therefore we collected observational data of sub-micron aerosol number size distributions from open databases (EBAS and ARM) and performed a standardized NPF analysis (see e.g. Kulmala et al., 2012) for these data. This way,

we were able to create an internally consistent data set on atmospheric NPF. This feature is not only crucial to the reliability of the result presented here, but also extremely beneficial for any further use of our data.

## 2    Description of the data and data analysis methods

Data of aerosol number concentration size distributions was obtained from the EBAS

(http://ebas.nilu.no/) and ARM (http://www.archive.arm.gov/discovery/) databases, and from several research groups running long-term atmospheric aerosol measurements. Mobility-based particle spectrometers (Differential Mobility Particle Sizer, DMPS; Scanning Mobility Particle Sizer, SMPS) typically have lower detection limits varying between 3 and 10 nm in particle diameter. In order to have comparable results between different sites, a common size range of

10–25 nm was used for nucleation mode particles in this study.

As part of the data analysis of this study, all the data was visually examined. Time periods when there was suspicion of instrument malfunction or other effects affecting the quality of the data were left out of the subsequent analysis. It should be noted, however, that the different measurement setups used at different measurement sites (and possibly changes in the

measurement setups) could introduce biases between the data sets from different measurement sites. In the literature, there exist a few guidelines for ambient aerosol size-distribution measurements and quality assurance procedures (e.g. Wiedensohler et al., 2012, 2017), but not all of the measurement sites follow these.

Altogether, we identified 36 measurement sites worldwide, where particle number size

distributions have been measured for at least one year (either continuously or during separate campaigns covering a full annual cycle). These sites were divided into five groups based on their general environmental characteristics (Table 1, Figure 1), ranging from polar and other remote



areas with low anthropogenic influence to heavily-polluted mega-cities. While most of the sites included in this study are located in Europe, we have at least two measurement sites from every other continent except Antarctica (only one site) and Australia (no sites). The measurement period lengths range from one year at two sites to just over 20 years in the Finnish boreal forest site. Most of the sites had data available for 5–10 years (Table 1).

Concerning the global spatial representativeness of the data sets analyzed in this study, it should be kept in mind that we have considered only measurements from continental areas which cover 29% of the Earth's surface, the rest being the oceans. Although the emissions of nucleation precursors and condensing vapours from the sea are much smaller than from the land vegetation (Carpenter et al., 2012), the larger overall surface area that they represent and the subsequent impacts on cloud cover may have significant influence on global climate. However, currently there is no evidence in the published literature from available measurements that NPF over the ocean is a common phenomenon compared to continental environments. Thus, as a future challenge, it would be very important to obtain similar long-term observations from at least few locations on the Atlantic, Pacific and Arctic Ocean.

## 2.1 Description of the measurement sites

Here we present a very short summary of the 36 sites included in this study. For more detailed information about each site, including their infrastructure, measurement program and environmental characteristics, we refer to the publications cited below.

### 2.1.1 Polar sites

The Zeppelin observatory (ZPL) is located on top of Mt Zeppelin, Svalbard (78° 56' N, 11° 53' E, 474 m above sea level (a.s.l.)), and is situated just outside the small community of Ny Ålesund. It is part of ACTRIS, GAW and ICOS programmes. The station is mostly unaffected by local sources and is considered to be within boundary layer most of the time. The station represents remote Arctic conditions, and offers a unique possibility to study the characteristic features of Arctic atmospheric trace constituents such as trace gases and aerosols (Tunved et al., 2013).

The Dome-C site (DMC) is located at the East Antarctica plateau at the Italian-French Concordia station, 1100 km away from the coast (75° 06' S, 123° 23' E, 3200 m a.s.l.; Järvinen et al., 2013). The station buildings are 1 km from the sampling site and upwind relative to the prevailing wind direction. The aerosol measurements with respect to the wind direction from the station are excluded from our analysis.

Alert station (ALE) of the Canadian Aerosol Baseline Measurement Program is the northernmost atmospheric measurement site in the world, located on the northeastern part of Ellesmere Island in Nunavut (82° 28' N, 62° 30' W, 75 m a.s.l.; Leaitch et al., 2013). It is part of the World Meteorological Organization's Global Atmosphere Watch (GAW) network. Alert is characterized by clean Arctic air during summer and long-range transport of more polluted air in southerly air masses primarily from Europe and Asia during winter and spring.

### 2.1.2 High-altitude sites

Jungfraujoch (JFJ) is a background site located in the Alps on a mountain ridge away from major pollution sources, and belongs to the GAW network (46° 33' N, 7° 59' E, 3580 m a.s.l;





Boulon et al., 2010; Bianchi et al., 2016; Bukowiecki et al., 2016). It contributes to numerous
networks, including GAW, ACTRIS, ICOS, NDACC, and AGAGE. For approximately 40% of
the time the station is inside clouds, and part of the time the station is considered to be in the free
troposphere.

The Puy de Dôme GAW research station (PDD; 45° 46' N, 2° 57' E) is located in central
France, approximately 200 km from the Atlantic Ocean coast and 150 km from the
Mediterranean Sea. It is on top of a volcano, at 1465 m above the sea level, and may be located
either in the continental boundary layer or the free troposphere (Venzac et al. 2009; Boulon et
al., 2011). It is representative of a large regional fingerprint and is classified as a background
regional site (Asmi et al., 2011).

Pico Espejo (PIC; 8° 30' N, 71° 6' W) is a tropical high altitude station located at 7° N on top of
the Venezuelan Andes at an altitude 4775 m above sea level (Schmeissner et al., 2011). It is
representative of the tropical free troposphere and for studies on the influence of orographic
lifting of boundary layer air to free troposphere.

Mukteshwar station (MUK; 29°26' N, 79°37' E, 2180 m a.s.l.) is located in northeast India
about 250 km from Delhi at the foothills of the central Himalayan mountains (Hyvärinen et al.,
2009; Neitola et al., 2011). The area surrounding the site consists of low mountains (peaks at
1500−2500 m a.s.l.) between the plains (100−200 m a.s.l.) and the Himalayas (peaks at 6000−
8000 m a.s.l). The site is influenced by regional polluted air that has been transported from the
plains below.

2.1.3   Remote sites

The Finokalia station (FKL; 35.3° N, 25.7° E; 235 m a.s.l) is located at the top of a hill over the
coastline, in the north east part of the island of Crete (Greece). The station is little influenced by
local anthropogenic sources and it is considered representative for the background marine
conditions of Eastern Mediterranean (Mihalopoulos et al., 1997).

Mace Head (MHD; 53.2° N, 9.8° W; 10 m a.s.l) is a coastal station on the west coast of Ireland
and receives clean marine air masses from the North East Atlantic approximately 50% of the
time. It serves as an excellent background marine aerosol characterization station as well as a
polluted European outflow station. Mace Head is a WMO-GAW global station, an EMEP
supersite and contributes to the ACTRIS and AGAGE networks. A full description can be found
in O'Connor et al., (2008) and O'Dowd et al, (2014).

Värriö (VÄR) SMEAR I (Station for Measuring Forest Ecosystem-Atmosphere Relations)
measurement site is located on top of the Kotovaara fjeld, surrounded by a 60-year old Scots
pine forest (67° 45' N, 29° 36' E, 390 m a.s.l). The station is close to the Finnish-Russian border
and is at times impacted by the air pollution coming from the Kola peninsula mining and
industrial areas 200-300 km north-east and east from the station (Kyrö et al., 2014).

The Pallas Atmosphere-Ecosystem Supersite station (PAL; 67°58' N, 24°07' E; 565 m a.s.l.) is
located in northern Finland. The main station building is within a natural park area, on top of a
hill above the tree line (Hatakka et al., 2003; Lohila et al., 2015). It is surrounded by vegetation
of low vascular plants, moss and lichen. The environment is representative of remote sub-Arctic
and boreal forests. The station is contributing to numerous European and global research
programmes, such as GAW, ICOS, ACTRIS and EMEP.





The Abisko measurement station (ABI) is located in a discontinuous permafrost zone at the Stordalen mire, approximately 14 km east of the small village of Abisko in northern Sweden (68.35°N, 19.05°E, 380 m a.s.l). The area is characterized by subarctic birch forest, wetlands and tundra ecosystems as well as a low population density (Svenningsson et al., 2008).

Tiksi Hydrometeorological Observatory (TKS; 71°36' N, 128°53' E, 10 m a.s.l) is located in northern Siberia on the coast of the Laptev Sea (Uttal et al., 2013; Asmi et al., 2016). The station is about 5 km southwest from the city of Tiksi, and about 500 m apart from the sea. The site is surrounded by low tundra vegetation with no trees.

The Waliguan Baseline Observatory (WLG; 36°17' N, 100°54' E, 3816 m a.s.l; Kivekäs et al.,
2009) is part of the GAW network, situated on top of Mt. Waliguan, located at the edge of northeastern part of the Qinghai-Xizang (Tibet) Plateau in a remote region of western China. Eventhough the station is located at a mountain peak and at very high altitude, a clear planetary boundary-layer–free-troposphere daily cycle in aerosol properties is not observed there. Therefore the Waliguan site is more representative of remote conditions.

### 2.1.4  Rural sites

Hyytiälä measurement site (HYY) is at the SMEAR II station located in Southern Finland 60 km north-east from Tampere (61°51'N, 24°17'E, 181 m a.s.l.; Hari and Kulmala, 2005). The station is equipped with extensive facilities to measure forest ecosystem-atmosphere interactions continuously and comprehensively. A rather homogeneous coniferous boreal forest surrounds
this rural continental station.

Aspvreten (ASP) is located ca. 2 km inland from the Baltic Sea (58.8°N, 17.4°E, 25 m a.s.l.), and some 80 km south of Stockholm. The surroundings are dominated by deciduous and coniferous forest, and the station is relatively unaffected from local anthropogenic activities (Tunved et al., 2004).

Preila station (PRL; 55.4°N, 21.0°E, 10 m a.s.l.) is located in the western part of Lithuania on the shore of the Baltic Sea, on the Curonian Spit. The Curonian Spit is a narrow sandy strip peninsula (0.4 to 4.0 km in width), which separates the Baltic sea from the Curonian Lagoon. Its width is approximately 2 km at the station Preila site. The dunes, up to 50 m height, as well as natural forests in low-lying lands predominate in the region. The marine, sub-marine climate is
specific to this terrain. This monitoring site was selected according to strict sitting criteria designed to avoid undue influence from point sources, area sources and local activities (Pauraite et al., 2015).

Tomsk Fonovaya Observatory (TMK) for monitoring atmospheric composition is located in the southern taiga belt of West Siberia (56°25' N, 84°4' E, 145 m a.s.l.; Matvienko et al., 2015). It is
200 representative of a background boreal environment and is situated on the bank of River Ob, 60 km west of the city of Tomsk. In close proximity to the site there is a mixed forest and large areas surrounding the site are covered mainly with coniferous trees.

Järvselja SMEAR-Estonia station (JRV) is located in the Järvselja Experimental Forest in the southeastern part of Estonia, about 35 km southeast of Tartu (56°16' N, 27°16' E, 36 m a.s.l.;
Noe et al., 2015; Vana et al., 2016). The site, located in the vicinity of Lake Peipus, is surrounded by mixed forest in the hemi-boreal forest zone. There are no large villages or cities near the site.



Hohenpeissenberg (HPB) is a GAW station located 60 km south of Munich on a mountain elevated 300 m above the surrounding countryside in southern Germany (47° 48' N, 11° 1' E, 988 m a.s.l.; Birmili et al., 2003). There are no major anthropogenic pollution sources nearby the station.

Vavihill station (VHL) is located at the southernmost part of Sweden (56° 1' N, 13° 9' E, 172 m a.s.l.; Kristensson et al., 2008). The station is away from local air pollution sources, but still within 40-45 km from the densely populated cities of Malmö and Copenhagen. Air masses arriving at the station from the direction of north-west to north-east are typically very clean.

K-puszta site (KPZ) is located in a rural area in Hungary, 15 km away from the nearest town of Kecskemet and 71 km from Budapest (46° 58' N, 19° 33' E, 125 m a.s.l.; Salma et al., 2016a). The station is in a clearing within a mixed forest of coniferous and deciduous trees.

Melpitz (MPZ) is located 40 km north-east of Leipzig, and surrounded by flat and semi-natural grasslands without any obstacles in all directions (51° 32' N, 12° 54' E, 87 m a.s.l.; Hamed et al., 2010). Agricultural pastures and wooded areas make up the wider regional surroundings of this regional background site. It is representative of the Central European background. Measurements at the Melpitz site are part of ACTRIS, GUAN, and GAW programs.

The San Pietro Capofiume station (SPC) is located in Po Valley, Italy, approximately 30 km from Bologna (44° 39' N, 11° 37' E, 11 m a.s.l.; Hamed et al., 2007). The Po Valley area is an industrial and agricultural area with high population density. The station itself is in rural area surrounded by the Adriatic Sea on the east and densely populated areas on its southern, western and northern sides.

The Cabauw (CBW) Experimental Site for Atmospheric Research (CESAR) is located in the central Netherlands close to the North Sea (51° 18' N, 4° 55' E, 60 m a.s.l.; Russchenberg et al., 2005). The CESAR observatory is located at a rural site with flat meadows at an otherwise densely populated area. It is representative for different environments depending on the wind directions.

The Harwell measurement site (HRW) is located in a rural environment in southern England (51° 34' N, 1° 19' W, 60 m a.s.l.; Charron et al., 2007). It is representative of the rural background in one of the more densely populated areas within Western Europe.

The Egbert site (EGB) of Environment and Climate Change Canada Centre for Atmospheric Research Experiments is located in rural Ontario surrounded by agricultural areas and small towns (44° 14' N, 79° 47' W, 251 m a.s.l; Rupakheti et al., 2005; Slowik et al., 2010; Pierce et al., 2014). With extensive forest to the north and a major urban center of Toronto about 80 km to the south, the site experiences many different types of aerosol depending on the wind direction.

The Southern Great Plains Central Facility site (SGP) of the US Department of Energy Atmospheric Radiation Measurement (ARM) program is located near Lamont, Oklahoma (36° 36' N, 97° 29' W, 300 m a.s.l.; Parworth et al., 2015). It is representative of the Great Plains region, and the surrounding areas have various anthropogenic activities including agriculture, animal husbandry, and oil and gas extraction.

Botsalano (BOT) is located in South Africa, 200 km west-northwest of Johannesburg in a game reserve in savannah environment (25° 32' S, 27° 75' E, 1400 m a.s.l.; Laakso et al., 2008;





Vakkari et al., 2011). Although there are no local anthropogenic sources, Botsalano is impacted
by aged emissions from the industrialized Highveld and is thus considered a semi-clean location.

Welgegund (WGD) is located in central South Africa within the grassland biome on a private
farm, with no local sources (26° 34' S, 26° 56' E, 1480 m a.s.l.; Tiitta et al., 2014; Jaars et al.,
2016). The site is impacted by the emissions from various strongly anthropogenically impacted
source regions (e.g. the Bushveld Complex 100 km to north and northeast, the Johannesburg-
Pretoria megacity and surrounding industries 100 km to the north and east, as well as the
Highveld and Vaal Triangle areas 100 km to east and southeast). It also has a wide clean sector
to the west. Welgegund is representative of the mosaic of grassland, cropland and anthropogenic
activities in the interior of southern Africa.

### 2.1.5   Urban and anthropogenically influenced sites

Marikana (MAR) is located in the middle of platinum group metal refineries near the city of
Rustenburg, South Africa (25° 42' S, 27° 29' E, 1170 m a.s.l.; Venter et al., 2012). In addition to
the industrial $SO_2$ emissions, the site is heavily impacted by domestic heating and cooking
emissions in nearby low-income residential areas.

The Helsinki measurement site (HEL) is the SMEAR III station in University of Helsinki
campus area (60° 12' N, 24° 58' E, 26 m a.s.l.; Hussein et al., 2008). The site is located next to a
busy road on a hill elevated by 20 m from the surrounding area.

The Beijing site (BEI) is located on a rooftop in the campus area of Peking University at
northwestern part of Beijing (40°00' N, 116°19' E, 50 m a.s.l.; Wu et al., 2007), as the Peking
University Urban Atmosphere Environment Monitoring Station (PKUERS). A major road is
located 500 m from the site, but there are no significant stationary air pollution sources nearby.

The Nanjing SORPES station (NAN) is located about 20 km northeast of downtown Nanjing,
China (32° 7' N, 118° 57' E, 25 m a.s.l.; Qi et al., 2015; Ding et al., 2016). With only few local
sources within its 2–3 km surroundings and generally upwind of the city, it can be considered as
a regional background site in the urbanized Yangtze River Delta region of Eastern China.

The measurements in Budapest (BUD) were conducted at two nearby sites: at the Budapest
Platform for Aerosol Research and Training in the city center on the bank of Danube (47° 29' N,
19° 4' E, 115 m a.s.l.; Salma et al., 2016b), and at the Konkoly Observatory in a near-city
background area (47° 30' N, 18° 58' E, 478 m a.s.l). The first of the sites is representative of
well-mixed urban air, and the second site is located in a wooded area (Nemeth and Salma,
2014).

The Sao Paulo measurement site (SPL) is located at the campus area of the University of Sao
Paulo 10 km from the city centre (23° 34' S, 46° 44' W, 750 m a.s.l.; Backman et al., 2012). The
Sao Paulo area is the world's 7th largest city, and the measurement site is representative of the
anthropogenic pollution of the city area with no strong local sources in the vicinity of the site.

## 2.2   Data analysis methods

All data sets were analyzed with the procedure following the particle number size distribution
data analysis guidelines presented by Kulmala et al. (2012). This was done in order to obtain a
dataset as coherent as possible. We classified every measurement day at each measurement site
into one of the following three categories: NPF event day, non-event day, or undefined day



(those days that could not be unambiguously classified into NPF or non-NPF days). We used the criteria originally introduced by Dal Maso et al. (2005), in which the class I event days are those during which the formation and subsequent growth of the nucleation mode particles is clearly distinguishable in the number size-distribution data for at least a few hours (Fig. 2). Class II event days are those during which there are evident inhomogeneities in the sampled air masses,
causing fluctuations in aerosol processes and in the observed particle size-distributions, but the regional NPF is still clearly observable. For a more detailed discussion of the analysis procedure, see Kulmala et al. (2012).

In order to quantify the intensity of individual NPF events, we calculated the formation rate $J_{nuc}$ of nucleation mode particles (10–25 nm in diameter) based on the following balance equation
(Kulmala et al., 2012):

$$J_{nuc} = \frac{dN_{nuc}}{dt} + \text{CoagS} \cdot N_{nuc} + \frac{\text{GR}}{\Delta d_{p,nuc}} \cdot N_{nuc} \tag{1}$$

Here $N_{nuc}$ is the total number concentration of 10–25 nm nucleation mode particles, CoagS is the coagulation sink due to the pre-existing larger particles, GR is the observed growth rate of particles through the 10–25 nm size range, and $\Delta d_{p,nuc}$ is the width of the 10–25 nm size range.
The growth rate GR was calculated by first fitting log-normal modes to the measured particle number size-distribution data using an automated algorithm developed by Hussein et al. (2008), and then following the time evolution of the geometric mean of the nucleation mode. A linear function was fitted to the data points of the nucleation mode size as function of time, and the slope of the fitted line gave the growth rate. The coagulation sinks were calculated based on the
dry size-distribution. The relative humidity dependent hygroscopic growth of the particles was not taken into account in our analyses, since this might differ between sites according to the particles' chemical composition and there are only few parameterizations for the hygroscopic growth available in the literature (Kulmala et al., 2012).

## 3   Results and discussion

Below we discuss three quantities that characterize atmospheric NPF events: the observed frequency of regional NPF events at individual measurement sites, the average formation rate of 10-25 nm particles ($J_{nuc}$) during each event, and the corresponding growth rate of 10−25 nm particles (GR). We will investigate both the overall behavior of these three quantities and their seasonal variability. Rather than looking at individual measurement sites, we will concentrate
our analysis on five groups of the sites that represent different environmental regimes: polar areas, high-altitude locations, remote areas, rural areas and urban areas. The individual values of the seasonal site specific medians of the NPF event frequencies, and nucleation mode particle formation and growth rates are given in Table 2. Note that the NPF frequency is the fraction of all class I and II NPF days from all the days with aerosol size-distribution data, but the particle
formation and growth rates are calculated only for the class I NPF events.

### 3.1   General characteristics of regional NPF and its seasonal cycle

Regional NPF events were observed at all the 36 sites throughout the year (Fig. 3), being most frequent at the three sites in Southern Africa (MAR, WGD, BOT) and least frequent at the two sites at high northern latitudes (ZPL, ALE). It should be noted that although at all the sites we



selected NPF events that exhibited formation and continuous growth of nucleation mode
        particles during several hours (i.e. fulfilling our criteria of regional NPF), the local conditions of
        each individual measurement site do influence the apparent NPF characteristics. For example, at
        high-altitude mountainous sites the orographic lifting of air parcels during the day can affect the
        conditions favourable to NPF. Such NPF events might show a temporal evolution of the particle
number size-distribution that is different from NPF events at locations with more homogeneous
        topography (Venzac et al., 2009). Thus, when comparing the results presented in this study to
        e.g. global modelling results, the regional representativeness should be kept in mind.

        The overall frequency of NPF did not show any consistent differences, or patterns, among the
        high-altitude, remote, rural and urban sites. There were, however, large site-to-site differences in
this frequency. Seasonally, the NPF frequency was typically the highest during March−May, the
        median value being equal to 31% among the seasonal-median values at each site. Since many of
        the northern hemisphere sites had very low NPF event frequencies during the local winter, the
        median value of this frequency was the lowest (8%) during the December−February period. The
        vast majority of the sites (30 out of 36) showed clearly more NPF events during the local spring
and summer compared with the local winter, as has also been reported in many previous studies
        in the literature (see e.g. Kulmala et al., 2012 and references therein).

        The observed formation rates of 10–25 nm particles increased, on average, with an increasing
        degree of anthropogenic influence, being one to two orders of magnitude higher in urban areas
        compared with most of the sites in remote and polar environments (Fig. 4) This indicates the
importance of anthropogenic vapors (such as sulphur dioxide, ammonia, amines) to NPF.
        Interestingly, the three high-altitude sites (JFJ, PDD, PIC) showed seasonal-median values of
        $J_{nuc}$ that were comparable to those at remote lower altitude areas. There are a few studies in
        which NPF has been studied in detail over different parts of the atmospheric column, and several
        mechanisms favoring or inhibiting NPF at different altitudes have been discussed without a clear
consensus (Crumeyrolle et al., 2010; Boulon et al., 2011; Rose et al., 2015). The seasonal
        variability of the particle formation rate was quite modest at most of the sites, and especially so
        when comparing it with the site-to-site differences in this quantity. The median value of $J_{nuc}$
        among the site-specific median values was between 0.4–0.6 cm$^{-3}$ s$^{-1}$ in all seasons. Its seasonal
        variation followed that of the NPF event frequency, except for December–February when NPF
event frequency was lowest but Jnuc values were similar to those in June–August.

        The observed growth rates of 10−25 nm particles were the lowest at the two northern high-
        latitude sites (ZPL, ALE; Fig. 5). Somewhat higher values of GR than the ones observed for
        northern sites, and with relatively minor site-to-site differences, were generally observed in
        remote and high-altitude sites. An exception to this pattern was PDD, which had clearly higher
values of GR than any other high-altitude or most of the remote sites. This has been observed to
        be caused by orographic vertical transport of particles nucleated in the boundary layer (Boulon
        et al., 2011). The particle growth rates tended to be the highest in rural and urban areas, even
        though large site-to-site differences were evident. The observed season-median values of GR
        varied from slightly below 1 nm/hour (DMC, spring) up to about 10 nm/hour at several sites
(e.g. EGB, BOT, WGD). Two rural stations Botsalano (BOT) and Welgegund (WGD) and urban
        station Marikana (MAR) located in South Africa showed similar seasonal variability of median
        GR, probably due to emissions of gaseous pollutants from various anthropogenically impacted
        source regions nearby. For most of the sites (33 out of 36), the season-median values of GR



were the highest during the local summer and the lowest during the local winter. As a result, the
overall median particle growth rate was clearly higher during the June−August period (4.0
nm/hour) compared with the December−February period (2.9 nm/hour). Exceptions are the three
South African stations (BOT, WGD, MAR), which showed considerably higher median GR
through the year (from September to May), except for the period June−August, when the median
GR values were comparable with other stations and more close to overall median GR. Also the
Egbert site (EGB) in Canada showed high median GR values (about 10 nm/hour) during the
period December–February, possibly due to increased anthropogenic impact during wintertime.

When looking at the seasonal variability of the three quantities discussed above, the observed
behavior of the particle growth rate is the easiest one to explain. Earlier studies based on
measurements in rural or remote locations have typically observed the highest values of GR
during the summer, and ascribed this feature to the higher emissions of biogenic aerosol
precursor compounds at higher ambient temperatures during the summer compared with other
season (Dal Maso et al., 2007; Pryor et al., 2010; Liao et al., 2014; Asmi et al., 2016). The
situation is more complicated in environments affected strongly by anthropogenic activities, e.g.
in practically all urban areas, where a large fraction of the compounds contributing to GR may
originate from anthropogenic precursors (e.g. Vakkari et al., 2015). Emissions of anthropogenic
aerosol precursor compounds may peak during any time of year, depending on human habits and
requirements influenced by weather and climate (e.g. heat and energy production), yet their
atmospheric oxidation to condensable vapors is expected to be strongest during summer in most
of the environments. It is likely that the strong atmospheric photochemistry, coupled with high
biogenic emissions of aerosol precursor vapors, largely explain the almost universal summer
maximum in GR at the sites considered here.

The NPF frequency had a clear summer-to-winter contrast similar to GR but, contrary to GR, it
peaked in March–May rather than in June–August at many of the sites. A regional modelling
study (Pietikäinen et al., 2014) indicated that the monthly average boundary layer burden of
freshly nucleated 3 nm particles (a quantity that depends on both the NPF event frequency and
particle formation rates) peaks in May-July in Europe. We find that the seasonal cycle of the
particle formation rate $J_{nuc}$ was rather weak for most of the sites, yet it appeared to follow
slightly better the seasonal cycle of the NPF frequency than that of GR. Several factors might
contribute to these differences. The most apparent of them are that, compared with GR, the
occurrence and strength of atmospheric NPF are expected to be more sensitive to the gas-phase
sulfuric acid concentration and pre-existing aerosol loading, and less sensitive to low-volatility
oxidation product concentrations of biogenic vapors (e.g. Westerveld et al., 2014; Dunne et al.,
2016). Furthermore, the value of $J_{nuc}$ is affected not only by the strength of NPF, but also by the
GR of particles starting from the nanometer size as well as the pre-existing aerosol load
affecting the coagulation sink (e.g. Lehtinen et al., 2007).

## 3.2   Relationships between the relevant quantities and implications

The annual-median particle formation rate and growth rate were positively correlated with each
other when considering all the 36 measurement sites together (Pearson correlation coefficient for
the logarithmic values is $r=0.72$, $p<0.01$), as well as for the sub-sets of high-altitude and rural
sites (Fig. 6). The other environments did not show such a relation, since in these environments
either the site-specific particle growth rates (at rural sites) or formation rates (at polar, high-



altitude and urban sites) had weak variability and were concentrated in a relatively narrow range of annual-median values. The positive relation between $J_{nuc}$ and GR was identifiable among the rural sites in all the seasons (results not shown here), and even among the remote sites during the spring and autumn.

On the annual basis, the particle formation and growth rates had a tendency to increase with increasing NPF event frequency between the different measurements sites (Fig. 6). A positive, yet moderate, correlation between $J_{nuc}$ and NPF event frequency was also observed when analyzing different seasons individually (results not shown here), as well as within the rural and remote sub-set of the sites. The relation between GR and NPF event frequency was rather weak, and remained so during the different seasons (results not shown here). None of the environments alone showed any sign of a relation between GR and NPF event frequency on an annual basis, but during summer a positive relationship was identifiable for the rural sub-set of the sites.

Intuitively, one would expect a certain degree of correlation between $J_{nuc}$, GR and NPF event frequency because higher values of all these quantities are favored by higher gas-phase production rates of low volatility vapors and by lower pre-existing aerosol loadings (e.g. Kulmala and Kerminen 2008; Westervelt et al., 2014). However, there are many other factors and processes that may cause a scatter in these relations. These factors and processes include the environmental and seasonal variability in i) the dominant new-particle formation mechanism (Kulmala et al., 2014; Dunne et al., 2016), ii) the availability of agents (ions, ammonia, amines, etc.) that are needed to stabilize molecular clusters containing sulfuric acid (Kirkby et al., 2011; Almeida et al., 2013; Schobesberger et al., 2015), iii) the mixture of compounds responsible for the main growth of newly-formed particles (see Vakkari et al., 2015, and references therein), and iv) meteorological conditions, which can indirectly influence the various processes and factors mentioned in (i), (ii) and (iii). In our data set there was considerable amount of scatter in each of the relationships between $J_{nuc}$, GR and NPF event frequency, which suggests that the values of these three quantities are affected by multiple factors with different degrees of importance among the individual locations.

In spite of the above discrepancies, the analysis of observed values of $J_{nuc}$, GR and NPF event frequency allowed us to make certain general statements on the importance of regional NPF. We need to keep in mind that regional NPF events considered in this study typically last at least for a few hours and, and as discussed earlier, that particles in the size range 10–25 nm in diameter are not very susceptible to coagulation and other loss processes. First, increases in the number concentration of particles larger than 10 nm due to a single NPF event are expected to be in the range from a few hundred to a few thousand particles cm$^{-3}$ per event at remote locations, and in the range from a few thousand up to more than $10^5$ particles cm$^{-3}$ per event in rural and urban locations, respectively. If these numbers are combined with the observed NPF event frequencies, and compared with total particle number concentrations measured in different types of environments (see, e.g. Asmi et al., 2013), it becomes clear that regional NPF is capable of explaining a dominant fraction of the total particle number concentration in both remote and polluted continental locations. This dominance may persist throughout the year in some of the locations, while being restricted to 1−3 seasons in some other locations. In different urban environments, there has been shown to be considerable variation in the contribution of NPF to the total particle number (Reche et al., 2011; Beddows et al., 2015). Second, depending on the location and season, we may estimate that it typically takes from a few hours to a couple of days



for the newly-formed particles to reach sizes larger than 50−100 nm in diameter at which they may act as CCN (see e.g. Kerminen et al., 2012). Our data suggests that in remote and rural locations, atmospheric CCN production associated with NPF tends to be most effective during summer and least effective during winter. Urban locations do not show any consistent seasonal pattern in this respect. Third, although regional NPF and the subsequent particle growth appear to be rather weak in polar areas during most of the year, the overall importance of atmospheric NPF for aerosol concentrations in polar areas is difficult to estimate based on our data. This is partly due to the limited number of continuous measurements available from polar sites, and partly because of the challenges in capturing atmospheric NPF that either have very low particle formation and growth rates or have overall characteristics that considerably differ from those in lower-latitude continental locations. Furthermore, polar and remote locations typically have lower concentrations of CCN-sized particles than anthropogenically influenced urban areas, thus the climatic importance of NPF cannot be evaluated only based on NPF frequency and particle formation and growth rates. In a recent modelling study, NPF influenced by ammonia emissions from a seabird-colony was shown to significantly contribute cooling in the Arctic area (Croft et al., 2016).

## 4    Summary and conclusions

By collecting a database on continuous particle number size distribution measurements at 36 continental sites worldwide, we investigated the overall and seasonal behavior of regional new particle formation in five different environmental regimes ranging from polar areas and remote sites to heavily-polluted megacities.

We found regional NPF events to take place at all the measurement sites throughout the year, with the exception of December–February at the sites at high latitudes (ZPL, ALE, ABI and TKS). NPF was most common (median of site-median NPF frequencies of about 30%) during the northern hemisphere spring and least common (less than 10%) during winter. No clear spatial pattern in the frequency of NPF according to environment type was observed, except that NPF events seemed to be most rare in polar areas during most seasons. We found that the formation rates of 10–25 nm particles ($J_{nuc}$) during the NPF events have a tendency to increase with an increasing degree of anthropogenic influence, being one to two orders of magnitude higher in urban areas compared with most of the remote and polar sites. The seasonal variability of $J_{nuc}$ was quite modest at most of the sites. We did not find any systematic environmental pattern for the growth rate (GR) of 10−25 nm particles during the NPF events, except that the GR were overall lowest in the polar regions. For the vast majority of the sites, the seasonal-median values of GR were the highest during the local summer and the lowest during the local winter. The observed values of $J_{nuc}$, GR and NPF indicate that regional NPF can explain a dominant fraction of the total particle number concentration, and give an important contribution to the cloud condensation nuclei population, at both remote and heavily-polluted continental locations.

We found that the connection between $J_{nuc}$, GR and NPF event frequency was at best moderate between the different measurement sites, as well as between the sites belonging to a certain environmental regime. The apparent lack of a strong relation between these three quantities is understandable due to the environmental and seasonal variability in the dominant new-particle formation mechanisms, in the abundances of compounds contributing to the initial steps of NPF



and subsequent particle growth, and in the prevailing meteorological conditions. For future studies, it would be very valuable to make detailed investigations on the interdependencies between $J_{nuc}$, GR and NPF event frequency, both at single measurement sites and between sites of seemingly similar environmental characteristics.

The data derived here will be helpful in evaluating, and possibly also in constraining, regional and large-scale atmospheric models that simulate aerosol formation and dynamics. However, it is also clear that more data similar to that presented in this study will be needed to better understand atmospheric NPF and its regional importance. Of specific importance in this respect are different urban areas in practically all over the world, additional remote and rural locations in Northern America, Asia and most of the Southern Hemisphere, and locations in polar areas. Furthermore, expanding the continental observations presented in this study to at least a few locations on the oceans covering 71% of the Earth's surface are needed for a comprehensive understanding of the global aerosol system and its effects on the global climate. For purely modeling purposes, or for the complementary use of models and *in situ* field and satellite measurements, it is probably sufficient to have particle number size distribution data down to a few nanometers (maximum 10 nm) in particle diameter. For a better understanding on NPF in different environments and comparison to corresponding laboratory data, such data should preferably be extended down to 1.5−3 nm in particle diameter  and ideally be complemented by measurements of the chemical composition of the growing clusters.





**Table and Figure captions**

**Table 1.** List of the measurement sites included in this study, the station name abbreviation used
to identify the sites in all the figures, station environment type, coordinates and altitude above
sea level (a.s.l.), time period from which data was analyzed, instrumentation, and the particle
size range. The color scheme in the first column represents the classification of the sites into
polar, high altitude, remote, rural and urban environments. The instruments used to measure
aerosol number size-distributions were Differential Mobility Particle Sizer (DMPS), Scanning
Mobility Particle Sizer (SMPS), Diffusion Particle Spectrometer (DPS) and Electrical Aerosol
Spectrometer (EAS).

**Table 2.** Site specific seasonal median values of NPF event frequencies (fraction of class I and
II NPF days from all the days with measurement data), and nucleation mode particle formation
and growth rates. A value is not given (indicated by –) if there were less than three quantifiable
NPF events at any given season.

**Figure 1.** Geographical coverage of the measurement sites offering long-term (at least one full
year) of aerosol number size distribution in sub-micron size range. The color of the points refer
to the color code in Table 1 used to group the sites according to their environment type.

**Figure 2.** An example of a new particle formation event observed in Hyytiälä, Finland, 15–16
March 2011, illustrating the continuous growth of the newly-formed aerosol particles for about
25 hours. The geometric mean size of the fitted log-normal size distributions are shown with
black dots, and the black dashed lines show the 10–25 nm size range that is used for calculating
the formation rate $J_{nuc}$ and growth rate $GR_{nuc}$.

**Figure 3.** Annual-median (a) and seasonal-median (b–e) frequency of the NPF formation events
at the different measurement sites. The dashed lines in panels (b–e) show the median seasonal
values, and the color scheme represents the classification of the sites into polar, high-altitude,
remote, rural and urban environments.

**Figure 4.** Annual-median (a) and seasonal-median (b–e) particle formation rate at the different
measurement sites. The dashed lines in panels (b–e) show the median seasonal values, and the
color scheme represents the classification of the sites into polar, high-altitude, remote, rural and
urban environments.

**Figure 5.** Annual-median (a) and seasonal-median (b–e) particle growth rate at the different
measurement sites. The dashed lines in panels (b–e) show the median seasonal values, and the
color scheme represents the classification of the sites into polar, high-altitude, remote, rural and
urban environments.

**Figure 6.** Annual-median, site-specific particle formation rate as a function of the corresponding
growth rate. The marker size is proportional to the annual-median NPF frequency and the
marker colors show the environment types of the sites.





**Table 1**

560

| | Station name and abbreviation | | Environment | Coordinates | Altitude (m a.s.l.) | Time period | Instrument | Size range (nm) |
|---|---|---|---|---|---|---|---|---|
| 1 | Mt. Zeppelin, Norway | ZPL | polar | 78° 56' N, 11° 53' E | 474 | 2005–2013 | DMPS | 10–800 |
| 2 | Dome-C, Antarctica | DMC | polar | 75° 6' S, 123° 23' E | 3200 | 2007–2009 | DMPS | 10–620 |
| 3 | Alert, Canada | ALE | polar | 82° 28' N, 62° 30' W | 75 | 2012–2014 | SMPS | 10–470 |
| 4 | Jungfraujoch, Switzerland | JFJ | high-altitude | 46° 33' N, 7° 59' E | 3580 | 2008–2009 | SMPS | 12–820 |
| 5 | Puy de Dome, France | PDD | high-altitude | 45° 46' N, 2° 57' E | 1465 | 2008–2009 | SMPS | 3–1000 |
| 6 | Pico Espejo, Venezuela | PIC | high-altitude | 8° 30' N, 71° 6' W | 4775 | 2007–2009 | DMPS | 10–470 |
| 7 | Mukteshwar, India | MUK | high-altitude | 29° 26' N, 79° 37' E | 2180 | 2005–2014 | DMPS | 10–750 |
| 8 | Mt. Waliguan, China | WLG | remote | 36° 17' N, 100° 54' E | 3816 | 2005–2007 | DMPS | 10–500 |
| 9 | Finokalia, Greece | FKL | remote | 35° 18' N, 25° 42' E | 235 | 2008–2012 | SMPS | 9–800 |
| 10 | Mace Head, Ireland | MHD | remote | 53° 12' N, 9° 48' W | 10 | 2005–2009 | SMPS | 8–470 |
| 11 | Värriö, Finland | VÄR | remote | 67° 45' N, 29° 36' E | 390 | 1997–2016 | DMPS | 3–860 |
| 12 | Pallas, Finland | PAL | remote | 67° 58' N, 24° 7' E | 565 | 2005–2014 | DMPS | 5–470 |
| 13 | Abisko, Sweden | ABI | remote | 68.35°N, 19.05°E | 380 | 2005–2007 | SMPS | 10–570 |
| 14 | Tiksi, Russia | TKS | remote | 71° 36' N, 128° 53' E | 10 | 2010–2012 | DMPS | 7–500 |
| 15 | Hyytiälä, Finland | HYY | rural | 61° 51' N, 24° 17' E | 181 | 1996–2016 | DMPS | 3–1000 |
| 16 | Aspvreten, Sweden | ASP | rural | 58° 48' N, 17° 24' E | 25 | 2006–2013 | DMPS | 10–470 |
| 17 | Preila, Lithuania | PRL | rural | 55° 24' N, 21° 0' E | 10 | 2009–2013 | SMPS | 8–850 |
| 18 | Tomsk, Russia | TMK | rural | 56°25' N, 84°4' E | 145 | 2011–2013 | DPS | 3–200 |
| 19 | Järvselja, Estonia | JRV | rural | 56° 16' N, 27° 16' E | 36 | 2012–2016 | EAS | 3–1000 |
| 20 | Hohenpeissenberg, Germany | HPB | rural | 47° 48' N, 11° 1' E | 988 | 2008–2015 | SMPS | 10–800 |
| 21 | Vavihill, Sweden | VHL | rural | 56° 1' N, 13° 9' E | 172 | 2008–2015 | DMPS | 3–900 |
| 22 | K-Puszta, Hungary | KPZ | rural | 46° 58' N, 19° 33' E | 125 | 2008–2014 | DMPS | 6–800 |
| 23 | Melpitz, Germany | MPZ | rural | 51° 32' N, 12° 54' E | 87 | 2008–2015 | DMPS | 5–800 |
| 24 | San Pietro Capofiume, Italy | SPC | rural | 44° 39' N, 11° 37' E | 11 | 2002–2016 | DMPS | 3–630 |
| 25 | Cabauw, Netherlands | CBW | rural | 51° 18' N, 4° 55' E | 60 | 2008–2009 | SMPS | 9–520 |
| 26 | Harwell, UK | HRW | rural | 51° 34' N, 1° 19' W | 60 | 2006 | SMPS | 12–440 |
| 27 | Egbert, Canada | EGB | rural | 44° 14' N, 79° 47' W | 251 | 2007–2008 | SMPS | 10–400 |
| 28 | Southern Great Plains, US | SGP | rural | 36° 36' N, 97° 29' W | 300 | 2011–2014 | DMPS | 12–740 |
| 29 | Botsalano, South Africa | BOT | rural | 25° 32' S, 27° 75' E | 1400 | 2006–2008 | DMPS | 11–840 |
| 30 | Welgegund, South Africa | WGD | rural | 26° 34' S, 26° 56' E | 1480 | 2010–2011 | DMPS | 11–840 |
| 31 | Marikana, South Africa | MAR | urban | 25° 42' S, 27° 29' E | 1170 | 2008–2010 | DMPS | 11–840 |
| 32 | Helsinki, Finland | HEL | urban | 60° 12' N, 24° 58' E | 26 | 2005–2016 | DMPS | 3–1000 |
| 33 | Beijing, China | BEI | urban | 40° 0' N, 116° 19' E | 50 | 2004 | DMPS | 3–1000 |
| 34 | Nanjing, China | NAN | urban | 32° 7' N, 118° 57' E | 25 | 2011–2013 | DMPS | 6–800 |
| 35 | Budapest, Hungary | BUD | urban | 47° 29' N, 19° 4' E | 115 | 2008–2012 | DMPS | 6–1000 |
| 36 | Sao Paulo, Brazil | SPL | urban | 23° 34' S, 46° 44' W | 750 | 2010–2011 | DMPS | 6–800 |



**Table 2**

| Site | Fraction of NPF days (%) | | | | Formation rate (cm-3 s-1) | | | | Growth rate (nm/h) | | | |
|------|---------|---------|---------|---------|---------|---------|---------|---------|---------|---------|---------|---------|
|      | Mar-May | Jun-Aug | Sep-Nov | Dec-Feb | Mar-May | Jun-Aug | Sep-Nov | Dec-Feb | Mar-May | Jun-Aug | Sep-Nov | Dec-Feb |
| ZPL  | 14.0 | 33.6 | 6.6  | 0.0  | 0.080 | 0.032 | 0.0066 | –     | 1.4 | 1.2 | 1.6 | –   |
| DMC  | 15.7 | 8.3  | 17.2 | 20.0 | 0.036 | –     | 0.0022 | 0.022 | 1.3 | –   | 0.5 | 2.5 |
| ALE  | 2.2  | 27.4 | 4.9  | 0.0  | 0.042 | 0.0081 | –     | –     | 0.8 | 1.1 | –   | –   |
| JFJ  | 23.9 | 9.7  | 13.7 | 3.9  | 0.035 | 0.042 | 0.052 | 0.043 | 2.7 | 3.1 | 1.5 | 3.0 |
| PDD  | 17.2 | 18.9 | 23.2 | 18.7 | 0.45  | 0.68  | 0.52  | 0.28  | 3.2 | 6.2 | 5.0 | 5.7 |
| PIC  | 17.6 | 13.8 | 18.1 | 31.9 | 0.24  | 0.049 | 0.24  | 0.14  | 2.7 | 3.0 | 4.0 | 4.0 |
| MUK  | 32.3 | 7.6  | 3.7  | 5.1  | 0.41  | 0.35  | 0.12  | 0.84  | 2.7 | 4.1 | 3.1 | 6.0 |
| WLG  | 23.7 | 20.7 | 25.5 | 24.6 | 1.7   | 1.0   | 0.48  | 1.1   | 2.4 | 5.1 | 1.4 | 2.2 |
| FKL  | 36.6 | 31.2 | 27.4 | 16.3 | 0.67  | 0.35  | 0.22  | 0.20  | 3.9 | 6.4 | 4.4 | 2.1 |
| MHD  | 29.3 | 17.3 | 10.0 | 6.5  | 0.31  | 0.49  | 0.41  | 0.35  | 2.1 | 2.8 | 2.7 | 2.3 |
| VÄR  | 27.8 | 16.8 | 11.8 | 4.8  | 0.11  | 0.10  | 0.060 | 0.038 | 1.9 | 3.9 | 2.4 | 2.2 |
| PAL  | 19.3 | 21.0 | 9.1  | 2.5  | 0.23  | 0.18  | 0.099 | 0.082 | 1.6 | 3.6 | 2.0 | 1.6 |
| ABI  | 14.0 | 33.5 | 15.3 | 0.0  | 0.37  | 0.13  | 0.034 | –     | 2.2 | 4.4 | 0.8 | –   |
| TKS  | 31.7 | 46.6 | 15.8 | 0.0  | 0.040 | 0.096 | 0.048 | –     | 2.7 | 3.4 | 2.3 | –   |
| HYY  | 47.2 | 22.2 | 19.9 | 7.4  | 0.52  | 0.21  | 0.37  | 0.29  | 2.2 | 4.6 | 2.8 | 1.9 |
| ASP  | 42.0 | 32.6 | 24.2 | 6.7  | 0.20  | 0.16  | 0.16  | 0.083 | 2.2 | 3.0 | 2.5 | 2.7 |
| PRL  | 16.8 | 15.3 | 15.5 | 3.9  | 0.67  | 0.097 | –     | 0.18  | 1.7 | 1.4 | –   | 3.3 |
| TMK  | 37.8 | 9.7  | 23.8 | 4.3  | 1.2   | 0.68  | 1.0   | 0.29  | 2.6 | 6.7 | 2.3 | 0.8 |
| JRV  | 39.1 | 9.6  | 18.8 | 4.7  | 0.76  | 1.3   | 0.48  | –     | 1.9 | 7.2 | 2.7 | –   |
| HPB  | 14.5 | 16.2 | 15.4 | 7.1  | 0.58  | 0.27  | 0.35  | 0.15  | 5.2 | 2.6 | 6.3 | 4.3 |
| VHL  | 58.8 | 58.0 | 41.0 | 12.2 | 0.63  | 0.88  | 0.23  | 0.15  | 3.3 | 3.1 | 2.4 | 3.4 |
| KPZ  | 32.0 | 23.6 | 40.8 | 18.8 | 1.5   | 1.7   | 1.6   | 1.1   | 3.6 | 4.0 | 3.7 | 3.3 |
| MPZ  | 45.0 | 57.6 | 19.3 | 6.5  | 2.7   | 1.8   | 0.69  | 0.80  | 2.5 | 2.7 | 2.5 | 2.6 |
| SPC  | 50.0 | 59.7 | 24.5 | 12.2 | 1.5   | 1.5   | 1.4   | 1.1   | 4.0 | 4.0 | 3.7 | 3.4 |
| CBW  | 31.1 | 39.2 | 21.3 | 16.4 | 0.97  | 1.2   | 1.0   | 0.79  | 3.9 | 4.9 | 3.5 | 2.9 |
| HRW  | 21.7 | 36.4 | 4.9  | 1.7  | 0.67  | 0.55  | 0.69  | 0.39  | 2.1 | 2.9 | 2.3 | 1.6 |
| EGB  | 66.3 | 47.6 | 56.4 | 17.9 | 0.92  | 0.73  | 0.94  | 1.3   | 6.0 | 6.1 | 5.4 | 9.6 |
| SGP  | 25.1 | 3.8  | 9.9  | 7.9  | 0.62  | –     | 0.96  | 0.39  | 4.0 | –   | 3.4 | 1.5 |
| BOT  | 75.6 | 70.7 | 59.3 | 73.9 | 3.1   | 2.6   | 5.3   | 3.9   | 7.5 | 7.2 | 10.9 | 9.9 |
| WGD  | 69.5 | 81.8 | 79.5 | 77.8 | 3.9   | 4.2   | 4.7   | 4.4   | 9.2 | 7.3 | 10.7 | 10.7 |
| MAR  | 76.4 | 63.6 | 60.3 | 76.7 | 4.9   | 3.2   | 4.9   | 4.8   | 8.1 | 6.1 | 8.5 | 9.7 |
| HEL  | 19.3 | 11.8 | 9.0  | 6.3  | 1.4   | 0.29  | 1.0   | 0.88  | 2.0 | 2.1 | 3.4 | 2.1 |
| BEI  | 78.0 | 44.7 | 60.5 | 58.2 | 8.4   | 6.3   | 5.9   | 5.9   | 3.3 | 4.6 | 2.0 | 1.6 |
| NAN  | 39.0 | 41.2 | 35.2 | 10.4 | 6.5   | 6.6   | 5.4   | 2.7   | 5.1 | 6.4 | 5.2 | 4.2 |
| BUD  | 42.3 | 28.7 | 28.0 | 13.6 | 0.97  | 0.78  | 0.9   | 0.55  | 4.6 | 5.1 | 4.5 | 2.9 |
| SPL  | 20.5 | 26.5 | 42.1 | 37.5 | 2.8   | 1.9   | 3.8   | 2.6   | 3.7 | 4.2 | 3.4 | 2.1 |





**Figure 1**

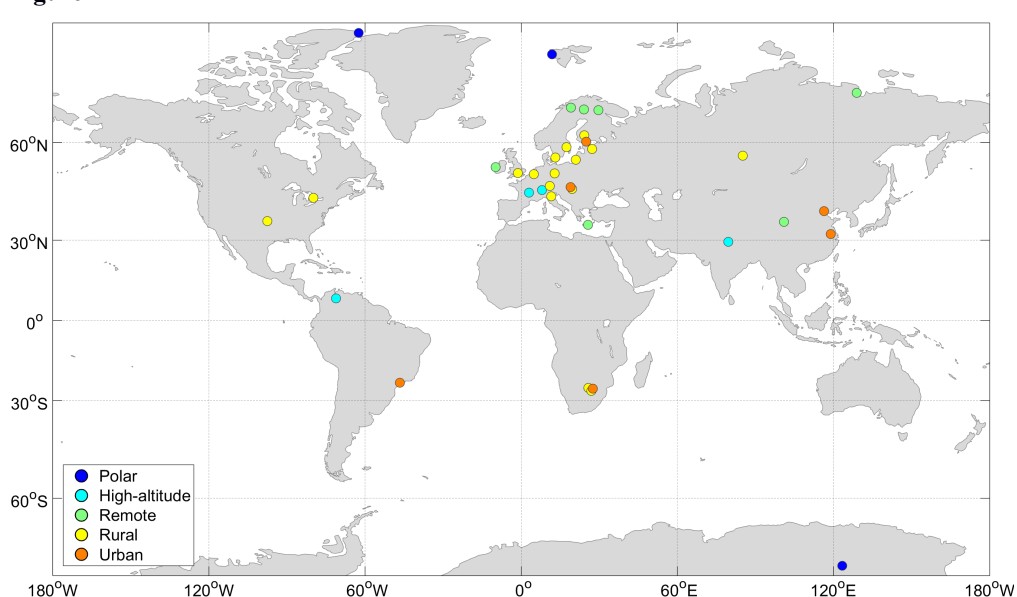




**Figure 2**

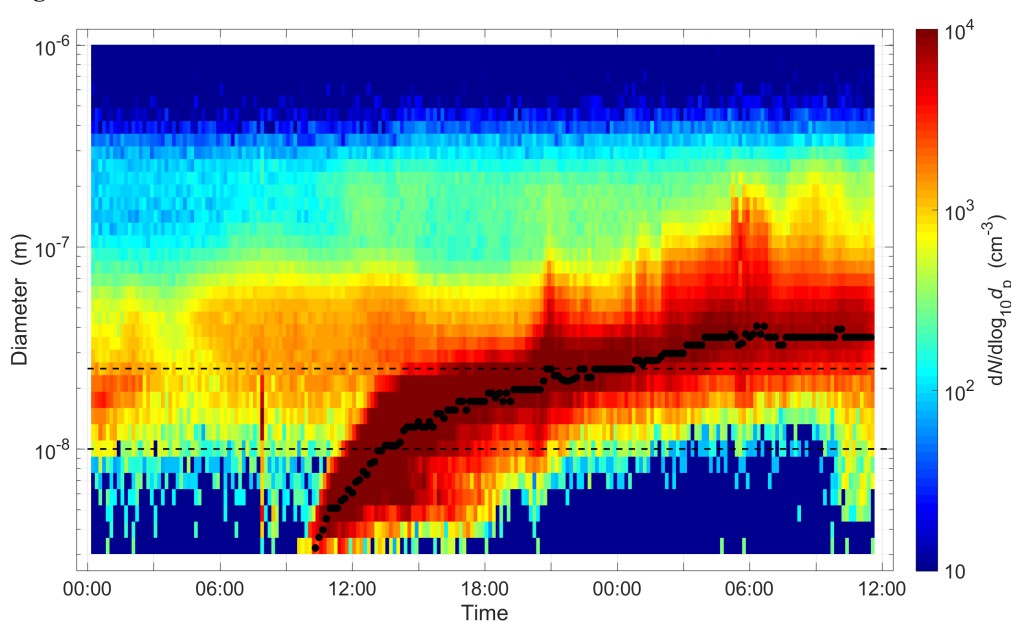





**Figure 3**

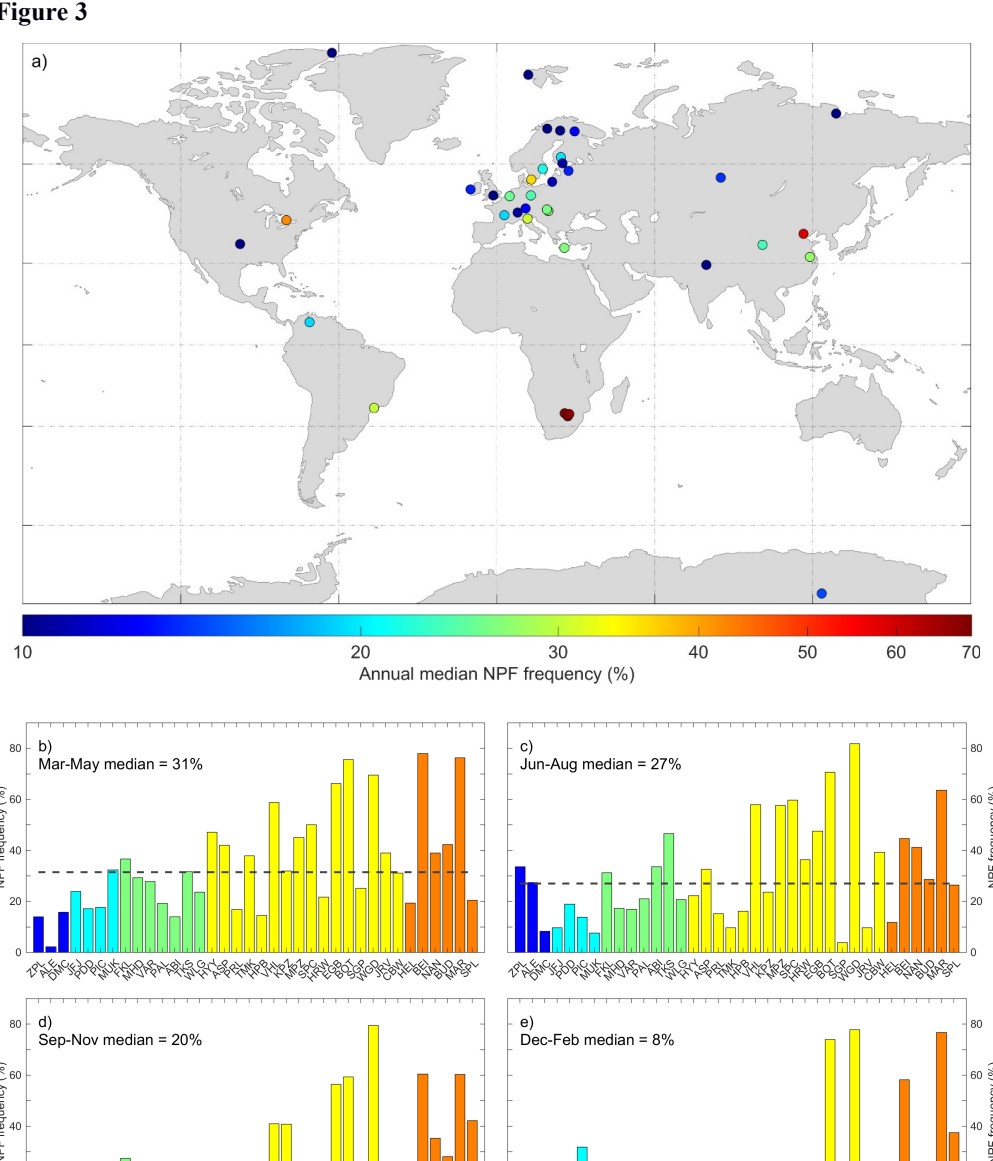



**Figure 4**







**Figure 5**

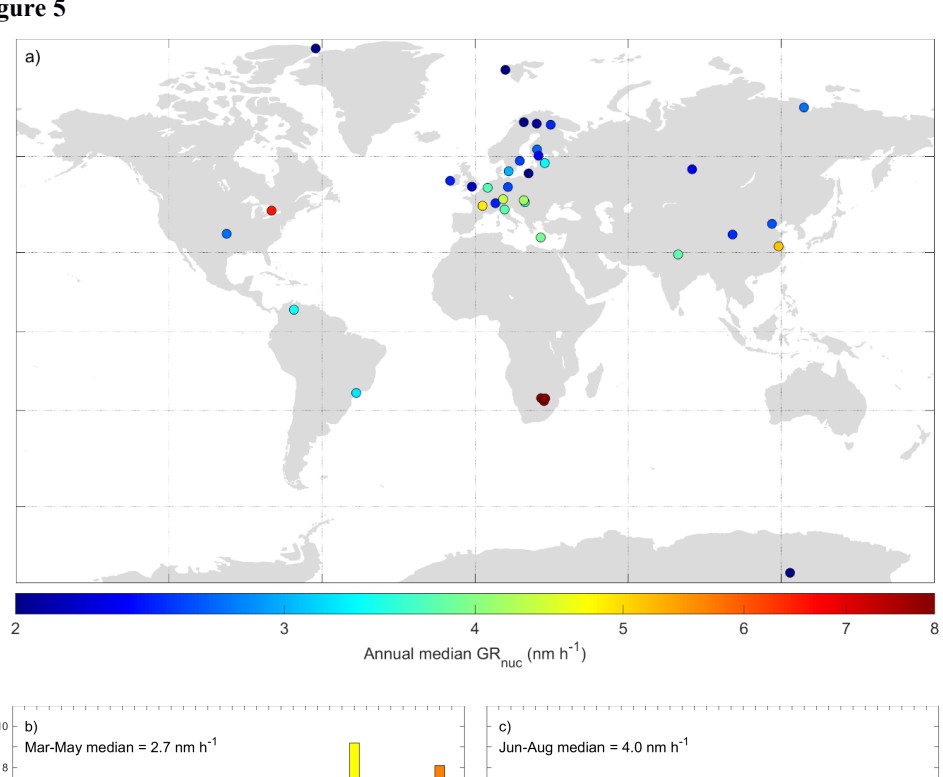

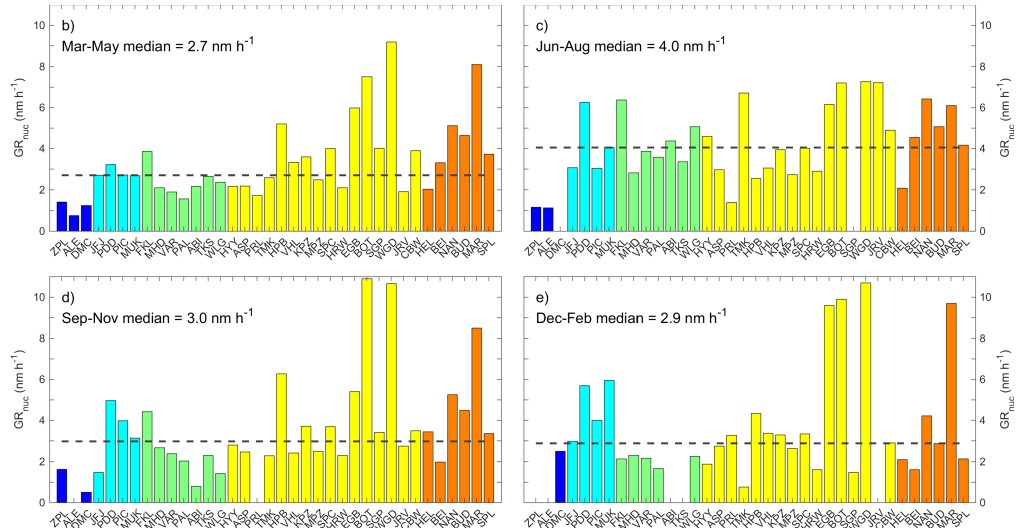





**Figure 6**

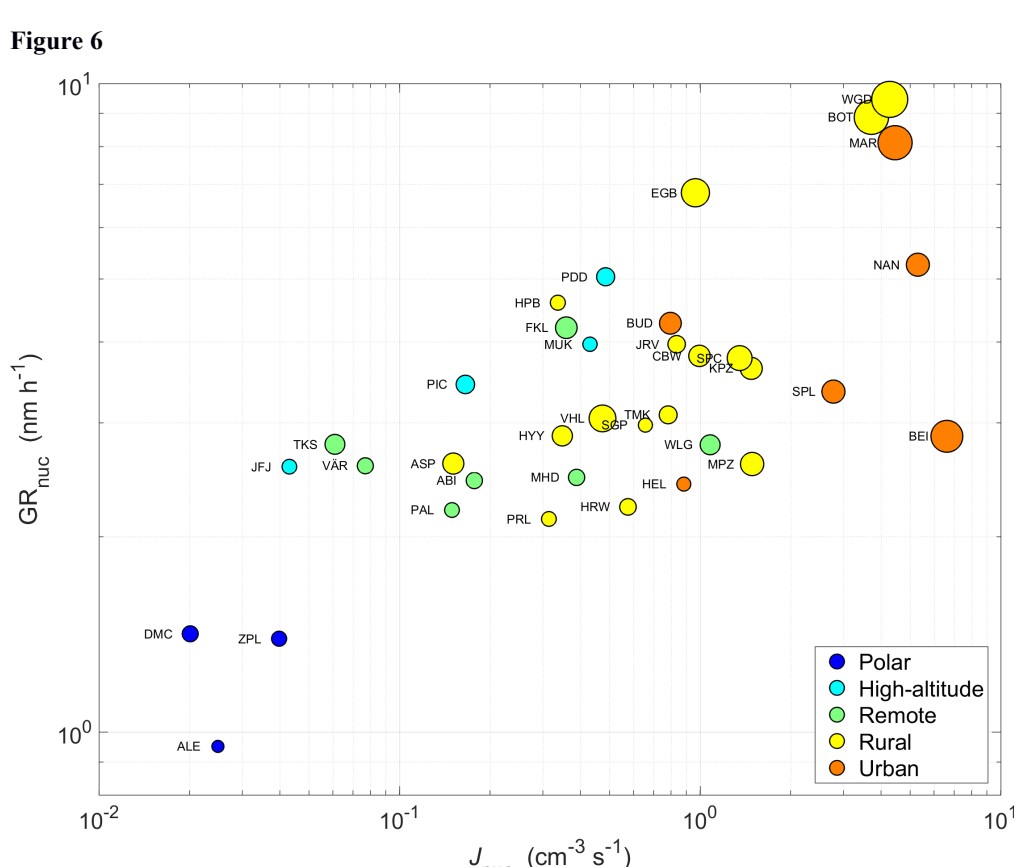





**Data availability**

The data sets analyzed in this study (NPF event frequencies, formation and growth rates) are available upon request from the corresponding author (tuomo.nieminen@uef.fi).

**Acknowledgements**

This work was supported by the Academy of Finland Center of Excellence program (project numbers 272041, 1118615, 307331), ACTRIS-2 under the European Union Research Infrastructure Action in the frame of the H2020 program for "Integrating and opening existing national and regional research infrastructures of European interest" (grant agreement 654109), EUSAAR (R113-CT-2006-026140) and EUCAARI (0136 833-2). MeteoSwiss is acknowledged for their long-term financial support since 1995 within the Swiss component of the Global Atmosphere Watch Programme of the World Meteorological Organization to the operations at Jungfraujoch site. The research at Zeppelin, Pico Espejo and Aspvreten stations has been carried out with the help of funding from the Swedish Research Council (Vetenskaprådet), Swedish Environmental Protection Agency (Naturvårdsverket) and Swedish development Cooperation Agency (SIDA). The measurements at Vavihill station are part of Swedish MERGE strategic research area. Continuous aerosol measurements at Melpitz site were supported by the German Federal Environment Ministry (BMU) grants F&E 370343200 ("Erfassung der Zahl feiner und ultrafeiner Partikel in der Außenluft") and F&E 371143232 ("Trendanalysen gesundheitsgefährdender Fein- und Ultrafeinstaubfraktionen unter Nutzung der im German Ultrafine Aerosol Network (GUAN) ermittelten Immissionsdaten durch Fortführung und Interpretation der Messreihen"). For Tiksi and Pallas sites, we acknowledge the funding from the Academy of Finland projects "Greenhouse gas, aerosol and albedo variations in the changing Arctic"(project number 269095) and "Novel Assessment of Black Carbon in the Eurasian Arctic: From Historical Concentrations and Sources to Future Climate Impacts" (NABCEA, project number 296302), and the funding from the European Union's Horizon 2020 programs under grant agreement No 727890 (INTAROS). Environment and Climate Change Canada is acknowledged for operating the Alert and Egbert sites. For Budapest site, the financial support by the National Research, Development and Innovation Office, Hungary (contracts K116788 and PD124283) is acknowledged. The measurements at Botsalano, Welgegund and Marikana sites have received funding from Academy of Finland projects "Air pollution in Southern Africa" (project number 117505) and "Atmospheric monitoring capacity building in Southern Africa" (project number 132640), from North-West University, and from Vilho, Yrjö and Kalle Väisälä Foundation. Measurements at Tomsk are carried out under support of the Department of Earth Sciences RAS. Measurements at Mukteshwar were performed with financial support by the Ministry of Foreign Affairs of Finland, Academy of Finland (264242, 268004, 284536), TEKES Finland, and DBT India (2634/31/2015). The Harwell measurement station was supported by the UK Department for Environment, Food and Rural Affairs. Institutional research funding IUT20-11 and IUT20-52 of the Estonian Ministry of Education and Research is acknowledged for the Järvselja site.

We acknowledge the following researchers for providing data from several stations: Nicolas Bukowiecki, Ernest Weingartner and Martine Collaud Coen (Jungfraujoch site); Thomas Tuch and Wolfram Birmili (Melpitz); Moa Sporre (Vavihill and Aspvreten), David Picard, Paolo Villani, Hervé Venzac and Paolo Laj (Puy de Dome); Giorgos Kouvarakis and Nikos Kalivitis (Finokalia); Dan Veber (Alert); Sander Mirme (Järvselja).



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
