# Peer review of "Global analysis of continental boundary layer new particle formation based on long-term measurements"

_Atmospheric Chemistry and Physics, 2018_

## Referee Comment (RC1) · Anonymous Referee #1 · 1 May 2018

This manuscript compiles and re-analyzes new particle formation and growth statistics from 36 surface sites. It is great to have all of these data in one places and analyzed homogeneously. I am very happy with the manuscript, and I only have a few minor comments. I feel it is ready to be published in ACP once these comments are addressed.

L6-7: Why, specifically, are only March-May and Dec-Feb being compared here? Is March-May the max and Dec-Feb the min? It would be good to say this, because now the choice seems arbitrary.

Throughout: "Westervelt" is misspelled as "Westerveld" is several places.

[Figure]

L88-97 and L509-516: Why is the free troposphere not mentioned here? Nucleation in the FT is hugely important for CCN (Merikanto et al., 2009).

L408-410: This sentence is strange. It's discussing the factors that determine Jnuc when Jnuc is inferred from dN(10-25nm)/dt, GR, and CoagSink; however, the sentence is written as if Jnuc *depends* on these values. Jnuc depends on vapor concentrations and temperature. It's only inferred using dN(10-25nm)/dt, GR, and CoagSink.

---

## Referee Comment (RC2) · Anonymous Referee #2 · 20 May 2018

General comments

This manuscript made a well understanding of atmospheric NPF and its regional importance based on the measurements at 36 continental sites around the world. It gathered a valuable dataset of aerosol number concentration size distribution and would be interesting to the readers of ACP. I recommend this manuscript for publication in ACP with minor revisions.

Table 1: Please add more information of each site for the data representativeness evaluation. For instance, the ratio of valid data determination of each site's measurements in this research.

[Figure]

L484-485: This sentence is ambiguous. "NPF was most common (median of site-median NPF frequencies ...and least common (less than 10%) during winter." According to the seasonal behavior of regional NPF contents in the manuscript, it would be better to replace "median of site-median NPF frequencies" with "site-median of seasonal-median NPF frequencies".
* * *

---

## Author Comment (AC1) · 13 Aug 2018

We thank the referee for the comments on our manuscript. Below we give our response to each of the comments and indicate changes made to the manuscript (referee's comments are shown in italics and our response in normal type).

*This manuscript compiles and re-analyzes new particle formation and growth statistics from 36 surface sites. It is great to have all of these data in one places and analyzed homogeneously. I am very happy with the manuscript, and I only have a few minor comments. I feel it is ready to be published in ACP once these comments are addressed.*

*L6-7: Why, specifically, are only March-May and Dec-Feb being compared here? Is March-May the max and Dec-Feb the min? It would be good to say this, because now the choice seems arbitrary.*

March-May was indeed the maximum in NPF occurrence frequencies and Dec-Feb minimum, averaged globally at these 36 measurement sites. As suggested by the referee, we have clarified this sentence in the revised manuscript: "We found that the NPF frequency has a strong seasonal variability. At the measurement sites analyzed in this study, NPF occurs most frequently in March-May (on about 30

*Throughout: "Westervelt" is misspelled as "Westerveld" is several places.*

We have corrected the spelling of the name throughout the revised manuscript.

*L88-97 and L509-516: Why is the free troposphere not mentioned here? Nucleation in the FT is hugely important for CCN (Merikanto et al., 2009).*

We agree with the referee that free-tropospheric nucleation is an important processes for the global aerosol number. The focus of this study, however, was to obtain a global picture of new particle formation in different environments. The long-term aerosol size-distribution data needed for this is only available from ground-based sites that are located inside the planetary boundary layer. Even the few high-altitude sites included in our study are only part-time in the free troposphere. For this reason we have not discussed nucleation in the FT in this manuscript.

*L408-410: This sentence is strange. It's discussing the factors that determine Jnuc when Jnuc is inferred from dN(10-25nm)/dt, GR, and CoagSink; however, the sentence is written as if Jnuc \*depends\* on these values. Jnuc depends on vapor concentrations and temperature. It's only inferred using dN(10-25nm)/dt, GR, and CoagSink.*

The referee is correct that $J_{nuc}$ (formation rate of 10-25 nm particles) is calculated from the measured N(10-25) using dN(10-25)/dt, GR (correcting for growth out of the 10-25 nm size range) and CoagSink (correcting for coagulation losses of 10-25 nm particles). However, ultimately $J_{nuc}$ depends on the actual nucleation rate $J^*$ (formation rate of $d^* \approx$ 1.5-2 nm clusters; Kulmala et al., 2013) and the losses occurring during the

condensational growth of these clusters to 10 nm size. As shown by Lehtinen et al. (2007), the dependence of $J_{\mathrm{nuc}}$ on $J^*$ can be expressed as

$$J_{\mathrm{nuc}} = J^* \exp(-\gamma \cdot d^* \cdot \frac{\mathrm{CoagS}}{\mathrm{GR}})$$

where the value of the parameter $\gamma$ depends on the size-distribution of the pre-existing aerosol. To clarify this sentence, we added to the revised manuscript: "This is because during the growth of the initial nucleated particles they are continuously scavenged by coagulation with the pre-existing aerosol."

Reference:
Kulmala, M. et al. (2013): Direct Observations of Atmospheric Aerosol Nucleation. Science 339, 943–946.

---

## Author Comment (AC2) · 13 Aug 2018

We thank the referee for the comments on our manuscript. Below we give our response to each of the comments and indicate the changes made to the manuscript (referee's comments are shown in italics and our response in normal type).

*General comments*
*This manuscript made a well understanding of atmospheric NPF and its regional impor-tance based on the measurements at 36 continental sites around the world. It gathered a valuable dataset of aerosol number concentration size distribution and would be in-teresting to the readers of ACP. I recommend this manuscript for publication in ACP*

*with minor revisions.*

*Table 1: Please add more information of each site for the data representativeness evaluation. For instance, the ratio of valid data determination of each site's measurements in this research.*

We added to Table 1 the percentage of days with available data between the start and end of the studied time period for each site. The modified Table 1 is shown below.

*L484-485: This sentence is ambiguous. "NPF was most common (median of site-median NPF frequencies . . .and least common (less than 10%) during winter." According to the seasonal behavior of regional NPF contents in the manuscript, it would be better to replace "median of site-median NPF frequencies" with "site-median of seasonal-median NPF frequencies".*

We agree with the referee: the NPF frequencies are indeed the median of all sites' seasonal-median values. We revised the sentence as "NPF was most common (site-median of seasonal-median NPF frequencies of about 30%) during the northern hemisphere spring and least common (less than 10%) during winter."
* * *
| | Station name and abbreviation | | Environment | Coordinates | Altitude (m a.s.l.) | Time period | Data coverage (%) | Instrument | Size range (nm) |
|---|---|---|---|---|---|---|---|---|---|
| 1 | Mt. Zeppelin, Norway | ZPL | polar | 78° 56' N, 11° 53' E | 474 | 2005–2013 | 91 | DMPS | 10–800 |
| 2 | Dome-C, Antarctica | DMC | polar | 75° 6' S, 123° 23' E | 3200 | 2007–2009 | 77 | DMPS | 10–620 |
| 3 | Alert, Canada | ALE | polar | 82° 28' N, 62° 30' W | 75 | 2012–2014 | 96 | SMPS | 10–470 |
| 4 | Jungfraujoch, Switzerland | JFJ | high-altitude | 46° 33' N, 7° 59' E | 3580 | 2008–2009 | 87 | SMPS | 12–820 |
| 5 | Puy de Dome, France | PDD | high-altitude | 45° 46' N, 2° 57' E | 1465 | 2008–2009 | 92 | SMPS | 3–1000 |
| 6 | Pico Espejo, Venezuela | PIC | high-altitude | 8° 30' N, 71° 6' W | 4775 | 2007–2009 | 86 | DMPS | 10–470 |
| 7 | Mukteshwar, India | MUK | high-altitude | 29° 26' N, 79° 37' E | 2180 | 2005–2014 | 87 | DMPS | 10–750 |
| 8 | Mt. Waliguan, China | WLG | remote | 36° 17' N, 100° 54' E | 3816 | 2005–2007 | 68 | DMPS | 10–500 |
| 9 | Finokalia, Greece | FKL | remote | 35° 18' N, 25° 42' E | 235 | 2008–2012 | 76 | SMPS | 9–800 |
| 10 | Mace Head, Ireland | MHD | remote | 53° 12' N, 9° 48' W | 10 | 2005–2009 | 87 | SMPS | 8–470 |
| 11 | Värriö, Finland | VÄR | remote | 67° 45' N, 29° 36' E | 390 | 1997–2016 | 94 | DMPS | 3–860 |
| 12 | Pallas, Finland | PAL | remote | 67° 58' N, 24° 7' E | 565 | 2005–2014 | 82 | DMPS | 5–470 |
| 13 | Abisko, Sweden | ABI | remote | 68.35°N, 19.05°E | 380 | 2005–2007 | 49 | SMPS | 10–570 |
| 14 | Tiksi, Russia | TKS | remote | 71° 36' N, 128° 53' E | 10 | 2010–2012 | 76 | DMPS | 7–500 |
| 15 | Hyytiälä, Finland | HYY | rural | 61° 51' N, 24° 17' E | 181 | 1996–2016 | 96 | DMPS | 3–1000 |
| 16 | Aspvreten, Sweden | ASP | rural | 58° 48' N, 17° 24' E | 25 | 2006–2013 | 94 | DMPS | 10–470 |
| 17 | Preila, Lithuania | PRL | rural | 55° 24' N, 21° 0' E | 10 | 2009–2013 | 59 | SMPS | 8–850 |
| 18 | Tomsk, Russia | TMK | rural | 56°25' N, 84°4' E | 145 | 2011–2013 | 92 | DPS | 3–200 |
| 19 | Järvselja, Estonia | JRV | rural | 56° 16' N, 27° 16' E | 36 | 2012–2016 | 79 | EAS | 3–1000 |
| 20 | Hohenpeissenberg, Germany | HPB | rural | 47° 48' N, 11° 1' E | 988 | 2008–2015 | 91 | SMPS | 10–800 |
| 21 | Vavihill, Sweden | VHL | rural | 56° 1' N, 13° 9' E | 172 | 2008–2015 | 84 | DMPS | 3–900 |
| 22 | K-Puszta, Hungary | KPZ | rural | 46° 58' N, 19° 33' E | 125 | 2008–2014 | 78 | DMPS | 6–800 |
| 23 | Melpitz, Germany | MPZ | rural | 51° 32' N, 12° 54' E | 87 | 2008–2015 | 87 | DMPS | 5–800 |
| 24 | San Pietro Capofiume, Italy | SPC | rural | 44° 39' N, 11° 37' E | 11 | 2002–2016 | 78 | DMPS | 3–630 |
| 25 | Cabauw, Netherlands | CBW | rural | 51° 18' N, 4° 55' E | 60 | 2008–2009 | 88 | SMPS | 9–520 |
| 26 | Harwell, UK | HRW | rural | 51° 34' N, 1° 19' W | 60 | 2006 | 86 | SMPS | 12–440 |
| 27 | Egbert, Canada | EGB | rural | 44° 14' N, 79° 47' W | 251 | 2007–2008 | 93 | SMPS | 10–400 |
| 28 | Southern Great Plains, US | SGP | rural | 36° 36' N, 97° 29' W | 300 | 2011–2014 | 91 | DMPS | 12–740 |
| 29 | Botsalano, South Africa | BOT | rural | 25° 32' S, 27° 75' E | 1400 | 2006–2008 | 80 | DMPS | 11–840 |
| 30 | Welgegund, South Africa | WGD | rural | 26° 34' S, 26° 56' E | 1480 | 2010–2011 | 97 | DMPS | 11–840 |
| 31 | Marikana, South Africa | MAR | urban | 25° 42' S, 27° 29' E | 1170 | 2008–2010 | 84 | DMPS | 11–840 |
| 32 | Helsinki, Finland | HEL | urban | 60° 12' N, 24° 58' E | 26 | 2005–2016 | 96 | DMPS | 3–1000 |
| 33 | Beijing, China | BEI | urban | 40° 0' N, 116° 19' E | 50 | 2004 | 61 | DMPS | 3–1000 |
| 34 | Nanjing, China | NAN | urban | 32° 7' N, 118° 57' E | 25 | 2011–2013 | 88 | DMPS | 6–800 |
| 35 | Budapest, Hungary | BUD | urban | 47° 29' N, 19° 4' E | 115 | 2008–2013 | 95 | DMPS | 6–1000 |
| 36 | Sao Paulo, Brazil | SPL | urban | 23° 34' S, 46° 44' W | 750 | 2010–2011 | 85 | DMPS | 6–800 |

**Fig. 1.** Modified Table 1 (column on data coverage added).

---

## Author Response (AR1)

**Response to reviewers' comments and revised version of manuscript acp-2018-304**

Below we give our responses to the reviewer's comments and indicate the changes made to the revised manuscript as a respond to the comments.

In addition to the changes listed below, we added one missing author's affiliation and three missing references to the reference list. A marked-up version of the manuscript showing the changes is at the end of this document.

**Reply to Anonymous Referee #1**

We thank the referee for the comments on our manuscript. Below we give our response to each of the comments and indicate the changes made to the manuscript (referee's comments are shown in italics and our response in normal type).

*This manuscript compiles and re-analyzes new particle formation and growth statistics from 36 surface sites. It is great to have all of these data in one places and analyzed homogeneously. I am very happy with the manuscript, and I only have a few minor comments. I feel it is ready to be published in ACP once these comments are addressed.*

*L6-7: Why, specifically, are only March-May and Dec-Feb being compared here? Is March-May the max and Dec-Feb the min? It would be good to say this, because now the choice seems arbitrary.*
March-May was indeed the maximum in NPF occurrence frequencies and Dec-Feb minimum, averaged globally at these 36 measurement sites. As suggested by the referee, we have clarified this sentence in the revised manuscript: "We found that the NPF frequency has a strong seasonal variability. At the measurement sites analyzed in this study, NPF occurs most frequently in March-May (on about 30% of the days) and least frequently in December-February (about 10% of the days)."

*Throughout: "Westervelt" is misspelled as "Westerveld" is several places.*
We have corrected the spelling of the name throughout the revised manuscript.

*L88-97 and L509-516: Why is the free troposphere not mentioned here? Nucleation in the FT is hugely important for CCN (Merikanto et al., 2009).*
We agree with the referee that free-tropospheric nucleation is an important processes for the global aerosol number. The focus of this study, however, was to obtain a global picture of new particle formation in different environments. The long-term aerosol size-distribution data needed for this is only available from ground-based sites that are located inside the planetary boundary layer. Even the few high-altitude sites included in our study are only part-time in the free troposphere. For this reason we have not discussed nucleation in the FT in this manuscript.

*L408-410: This sentence is strange. It's discussing the factors that determine Jnuc when Jnuc is inferred from dN(10-25nm)/dt, GR, and CoagSink; however, the sentence is written as if Jnuc \*depends\* on these values. Jnuc depends on vapor concentrations and temperature. It's only inferred using dN(10-25nm)/dt, GR, and CoagSink.*
The referee is correct that Jnuc (formation rate of 10-25 nm particles) is calculated from the measured N(10-25) using dN(10-25)/dt, GR (correcting for growth out of the 10-25 nm size range) and CoagSink (correcting for coagulation losses of 10-25 nm particles). However, ultimately Jnuc depends on the actual nucleation rate $J^*$ (formation rate of $d^* \approx 1.5\text{-}2$ nm

clusters; Kulmala et al., 2013) and the losses occurring during the condensational growth of these clusters to 10 nm size. As shown by Lehtinen et al. (2007), the dependence of Jnuc on J* can be expressed as

Jnuc = J*·exp(-γ·d*·CoagS/GR)

where the value of the parameter γ depends on the size-distribution of the pre-existing aerosol. To clarify this sentence, we added to the revised manuscript: "This is because during the growth of the initial nucleated particles they are continuously scavenged by coagulation with the pre-existing aerosol."

**Reply to Anonymous Referee #2**

We thank the referee for the comments on our manuscript. Below we give our response to each of the comments and indicate the changes made to the manuscript (referee's comments are shown in italics and our response in normal type).

*General comments*
*This manuscript made a well understanding of atmospheric NPF and its regional importance based on the measurements at 36 continental sites around the world. It gathered a valuable dataset of aerosol number concentration size distribution and would be interesting to the readers of ACP. I recommend this manuscript for publication in ACP with minor revisions.*

*Table 1: Please add more information of each site for the data representativeness evaluation. For instance, the ratio of valid data determination of each site's measurements in this research.*
We added to Table 1 the percentage of days with available data between the start and end of the studied time period for each site. The modified Table 1 is shown below.

*L484-485: This sentence is ambiguous. "NPF was most common (median of site-median NPF frequencies . . .and least common (less than 10%) during winter." According to the seasonal behavior of regional NPF contents in the manuscript, it would be better to replace "median of site-median NPF frequencies" with "site-median of seasonal-median NPF frequencies".*
We agree with the referee: the NPF frequencies are indeed the median of all sites' seasonal-median values. We revised the sentence as "
[revised manuscript text omitted]